**Data Availability Statement:** All relevant data are within the manuscript and its Supporting Information files.

# Trends in de-lousing of Norwegian farmed salmon from 2000–2019—Consumption of medicines, salmon louse resistance and non-medicinal control methods

**Elena Myhre Jensen**[1]*, **Tor Einar Horsberg**[1], **Sigmund Sevatdal**[2], **Kari Olli Helgesen**[3]

**1** Faculty of Veterinary Medicine, Sea Lice Research Center, Norwegian University of Life Sciences (NMBU), Oslo, Norway, **2** VESO Vikan, Namsos, Norway, **3** Norwegian Veterinary Institute, Oslo, Norway

\* elena.myhre.jensen@nmbu.no

## Abstract

The salmon louse *Lepeophtheirus salmonis* has been a substantial obstacle in Norwegian farming of Atlantic salmon for decades. With a limited selection of available medicines and frequent delousing treatments, resistance has emerged among salmon lice. Surveillance of salmon louse sensitivity has been in place since 2013, and consumption of medicines has been recorded since the early 80's. The peak year for salmon lice treatments was 2015, when 5.7 times as many tonnes of salmonids were treated compared to harvested. In recent years, non-medicinal methods of delousing farmed fish have been introduced to the industry. By utilizing data on the annual consumption of medicines, annual frequency of medicinal and non-medicinal treatments, the aim of the current study was to describe the causative factors behind salmon lice sensitivity in the years 2000–2019, measured through toxicity tests–bioassays. The sensitivity data from 2000–2012 demonstrate the early emergence of resistance in salmon lice along the Norwegian coast. Reduced sensitivity towards azamethiphos, deltamethrin and emamectin benzoate was evident from 2009, 2009 and 2007, respectively. The annual variation in medicine consumption and frequency of medicinal treatments correlated well with the evolution in salmon louse sensitivity. The patterns are similar, with a relatively small response delay from the decline in the consumption of medicines in Norway (2016 and onward) to the decline in measured resistance among salmon louse (2017 and onward). 2017 was the first year in which non-medicinal treatments outnumbered medicinal delousing treatments as well as the peak year in numbers of cleaner-fish deployed. This study highlights the significance of avoiding heavy reliance on a few substance groups to combat ectoparasites, this can be a potent catalyst for resistance evolution. Further, it demonstrates the importance of transparency in the global industry, which enables the industry to learn from poor choices in the past.

**Funding:** The study was financially supported by the Norwegian Research Council through the Centre for Innovation, Sea Lice Research Centre, grant number NFR 203513/O30. This funder had no role in study design, data collection and analysis, decision to publish, or preparation of the manuscript, and provided only financial support in the form of salaries to [EMJ]. One of the authors of the study [SS] is employed by the commercial company VESO, Oslo, Norway, and contributed to study design, data collection and preparation of the manuscript. The company VESO, however, had no role in study design, data collection and analysis, decision to publish or preparation of the manuscript. The specific role of each author is articulated in the 'author contributions' section.

**Competing interests:** No competing interests exist. One of the authors of the study [SS] is employed by the commercial company VESO, Oslo, Norway, and contributed to study design, data collection and preparation of the manuscript. The company VESO, however, had no role in study design, data collection and analysis, decision to publish or preparation of the manuscript. This does not alter our adherence to PLOS ONE policies on sharing data and materials.

## Introduction

Farming of Atlantic salmon has become one of Norway's largest businesses and farmed salmon the largest export from the country. Over one million tonnes of salmon at a value of 7.2 billion euros (converted from NOK 12.02.2020) was exported in 2019 [1]. One significant factor that complicates the production of salmon worldwide is the ectoparasite *Lepeophtheirus salmonis* (the salmon louse). According to the Intrafish Sea Lice Report 2019, annual costs associated with sea lice management were estimated at USD 525 million and USD 350 million in the 2 main markets, Norway and Chile [2]. In high numbers these parasites have the potential of critically wounding their salmonid hosts. In addition, salmon lice from farmed salmon can infest wild salmonids, thus compromising these already strained populations [3]. Regulations from the Norwegian Food Safety Authorities are therefore in place, ensuring that infestation levels are controlled. In 2008, the regulation stated that infection levels above 0.5 adult female lice per fish required de-lousing. In 2009, the regulation was adapted to state that exceeding 0.5 adult females per fish during warm months or 1 adult female louse per fish during cold months required de-lousing. The current limit, which was set in force in 2013, states that de-lousing measures must be deployed before levels reach 0.2 adult female lice per fish during spring or 0.5 adult female lice per fish the remainder of the year, and weekly (4° C or above) or biweekly (below 4° C) lice counts are mandatory [4]. The regulation is strictly enforced by the authorities.

Chemical de-lousing, applied topically or in feed, has since the early 1980s and up to 2015 been the dominant way of keeping infestation levels below the regulated limit and usage has been extensive. Records of utilization of different compounds in Norway are available from 1981, from the Norwegian Institute of Public Health [5]. Organophosphates (metrifonate and dichlorvos) was the only group used until 1993 when hydrogen peroxide became available as a treatment. Chitin synthesis inhibitors (diflubenzuron and teflubenzuron) and the pyrethroid cypermethrin became available in 1996, and emamectin benzoate in 1999. Since then, no new substances have been introduced for use against salmon lice. This medicinal dependence of few medicinal classes has led to the evolution of resistance in salmon lice and hence poor treatment efficacies [6–9].

Resistant parasites are suspected when there is a loss of treatment efficacy, and verified through bioassays or other laboratory assays. Bioassays are most commonly used and are toxicological assays where groups of parasites are exposed to different concentrations of the medicinal compound. The sensitivity level is evaluated from the survival following exposure. Before 2013, laboratory six-dose bioassays developed by Sevatdal & Horsberg [8] were conducted by the Norwegian School of Veterinary Science, the Veterinary Center for Contract Research (VESO) and some fish health services. By analysis of the dose-response curve, the population sensitivity was evaluated by determination of the median effective concentration of the medicinal compound, the $EC_{50}$ value. As a response to the emerging resistance, the Norwegian Food Safety Authority wanted to set up a coast-long surveillance program for resistance in Norway. The program was launched in 2013 [10]. The individual bioassays were to be conducted by local fish health services. It soon became clear that the six-dose assays were too complicated for field use on a larger scale. The bioassay protocols chosen were based on two-level bioassays plus a control group, where the lowest dose discriminates between fully sensitive parasites and parasites with reduced sensitivity, while the highest dose is predicting the treatment efficacy using the labelled dosage [11]. All available bioassay data, both six-dose assays and two-dose assays were compiled in connection with this study.

Since 2017, the number of treatments using non-medicinal methods of lowering infestation levels has overtaken the number of treatments using medicinal compounds [12]. Several non-medicinal approaches to de-lousing are available in Norway and other salmonid producing countries. Although safer for the environment and no detected resistance development so far,

some of these methods involve stressful crowding, pumping and other types of handling. Their impact on fish welfare have therefore been questioned. These consist of freshwater bathing, warm water dips, cold water bathing [13], use of lasers to kill individual lice on the fish, mechanical removal of parasites by soft brushes and/or high pressure pumps and deployment of cleanerfish with the farmed salmon. Overton et al. [14] compared reported mortality rates associated with medicinal and non-medicinal treatments and found that thermal operations caused greatest mortality increase, followed by mechanical treatments, hydrogen peroxide treatments and then treatments with azamethiphos, deltamethrin and cypermethrin. Preventive strategies such as synchronized fallow periods within production zones, synchronization of treatments, use of snorkel cages, functional feed, deep water feeding and plankton nets will not be addressed further in this study. In later years, concerns have been raised not only about fish wellbeing during some of these practices [13–16], but also that the lice may in fact adapt to these challenges as well [12, 17] as there may be genetic variation in susceptibility, and hence survival and onward input into the gene pool [18].

In this descriptive retrospective study, we aim to describe the development in salmon louse treatment frequency, medicine consumption and method choice over the last two decades and connect this to the sensitivity level of salmon lice as measured in bioassays conducted in the same period. By understanding past mistakes in Norwegian salmon farming, we can avoid the same resistance problems in the future as we face today.

## Materials and methods

### Data collection

The annual consumption of medicinal compounds for human and veterinary medicine are publicly available in Norway, published by the Norwegian Institute of Public Health. Information about consumption of anti-parasitic substances in Norwegian salmon aquaculture has been recorded since 1981 and was initially collected by the monopoly medicinal wholesaler Norwegian Medicinal Depot, later by the WHO Collaborating Centre for Drug Statistics Methodology at the Institute of Public Health [5]. The annual volume of salmonids harvested was collected from Statistics Norway [19].

A three-year EU funded project (SEARCH) aiming to establish the baseline sensitivity of salmon lice in Norway, Scotland, Ireland and Canada was initiated in 2000 [8]. The backbone for the project was development of six-dose bioassays (toxicological test on living parasites) where the most important parameter was the $EC_{50}$ values (median effective concentration–the concentration at which 50% of the test population are moribund or dead). These assays are referred to as "traditional bioassays". After the project period, Sevatdal and colleagues, as well as local fish health services, continued conducting bioassays as routine surveillance of sensitivity and also to advise fish health professionals on alternative treatments when they experienced inadequate treatment efficacies of a compound. The sampling and bioassay methods were described in Sevatdal & Horsberg [8], however the concentrations varied somewhat in later years to capture the increased resistance level. In order to demonstrate how the sensitivity level of salmon lice developed in the time period before the organized sensitivity surveillance commenced in 2013, $EC_{50}$ values from bioassays conducted in these years were compiled.

In 2013, a simplified and standardized two-dose bioassay was developed for the national sensitivity surveillance program in Norway [11]. The reports from the program are published annually [10]. The raw data from all bioassays conducted during the period 2013–2019 were obtained from the Norwegian Veterinary Institute.

As of 2012, data reported by production companies about medicinal and non-medicinal treatments taking place each week, in addition to other mandatory reported details, were

made publicly available through the portal Barentswatch [20]. A complete dataset of all reported treatments was downloaded from the site.

Lastly, information about the annual number of cleanerfish deployed in the production of Atlantic salmon and Rainbow trout has been recorded by the Norwegian Directorate of Fisheries since 1998. The complete dataset was downloaded from their website [21].

## Data sorting and filtering

**Traditional six-dose bioassays.** A total of 796 traditional bioassay results were compiled from the period 2000–2015. The following exclusion criteria were then applied:

- Traditional six-dose bioassays after 2012 were excluded since the number of two-dose assays greatly outnumbered these from 2013 (n = 49).

- Bioassays in which a combination of two (or more) substances were used in the same assay were excluded as the sensitivity level towards the individual compounds could not be determined (n = 12).

- One bioassay with only adult males was excluded (n = 1)

- Obvious outliers in the dataset which deviated substantially from the interquartile range (in bioassays with: azamethiphos $EC_{50}$ values > 500 ppb (n = 3), cypermethrin $EC_{50}$ values > 140 ppb (n = 1) and deltamethrin $EC_{50}$ values > 130 ppb (n = 2)) were excluded from further analysis.

After the exclusion process, we were left with a dataset containing **720** observations which were used in the analyses described below.

**Bioassays in national surveillance program.** A total of 1488 bioassay results from bioassays were compiled from the national sensitivity surveillance program for years 2013–2019. The following exclusion criteria were then applied:

- Bioassays in which mortality in the control group exceeded 20% were excluded (n = 65)

- Results from tests run with emamectin benzoate from year 2013 were excluded because the dose used in high dose group was different in the following years, and comparing these would introduce uncertainty (n = 37)

- Bioassays in which high dose groups are missing (NA) were excluded (n = 1)

After the exclusion process, we were left with a dataset containing **1385** observations.

**Reported treatment events.** A total of 45,788 measures to lower infestation levels on farmed salmon was reported in the period 2012–2019, and categorized as either "medicinal", "non-medicinal/mechanical" or "cleanerfish". Information on cleanerfish deployment were excluded in this dataset as it was presented in a more detailed fashion from the Norwegian Directorate of Fisheries (n = 23,362). Numbers of medicinal and non-medicinal treatments were isolated and sorted by year using the statistical software R and gave 22,426 observations that were analyzed further.

**Cleanerfish deployment.** The total number of cleanerfish individuals (in 1000) deployed in Norwegian salmonid farms from years 2000–2019 was isolated from the complete dataset and analyzed further.

## Data processing and analysis

**Consumption of medicines and slaughter volumes.** The total amount of fish treated with each delousing compound per year between 2000 and 2019 were for orally administered agents calculated from the labelled dosage (mg/kg). For bath treatments, the total amount of

fish treated per year was calculated assuming that 50 kg fish were treated per cubic meter treatment bath, using the labelled dosage for the product (mg/m$^3$).

**Traditional bioassays.** Data from traditional bioassays were subjected to probit modelling using the program PoloPlus (LeOra Software LLC) or the statistical software JMP Plus Pro 14.3.0 (SAS) with "Probit Simple V2" probit add-in. The median effective concentration, EC$_{50}$, was used. The EC$_{50}$ values from these bioassays (2000–2012) were grouped by substance and year, and presented as boxplots using the statistical software R [22] with the package "ggplot2".

**Bioassays from the surveillance program.** The results from bioassays conducted in 2013–2019 were obtained from the Norwegian Veterinary Institute, and survival of parasites at the concentrations applied in these assays were calculated by the following equation:

$$Survival = \frac{number\ of\ surviving\ in\ group}{total\ number\ in\ group} * 100$$

The statistical software R [22] with packages "ggplot2" and "egg" was used for further processing of data. The results were sorted on substance and date, and LOWESS curves with associated 95% confidence interval were fitted to the data.

**Reported treatment events.** Information about the frequency of medicinal and non-medicinal treatment events were collected from the open information source Barentswatch [20], compiled with R [22] with package "tidyr", and the results were presented in bar graphs.

**Deployment of cleanerfish.** The number of deployed cleanerfish individuals (in 1000) for the country as a whole was sorted by year and tabulated in Excel. The data was then presented as a simple line plot.

## Results

### Use of medicinal compounds against salmon lice

Medicinal compounds have been used against salmon lice since the early 1980s. Until 1993, the organophosphates metrifonate and dichlorvos were the only compounds used (S1 Table). At this time, the efficacy of these were lost in parts of the country and new treatments were introduced [23]: hydrogen peroxide, azamethiphos, diflubenzuron, teflubenzuron, pyrethrins, cypermethrin, deltamethrin and finally emamectin benzoate in 1999.

The use of the various agents in the period 2000–2012 and 2013–2019 are listed in Tables 1 and 2, respectively. Azamethiphos was used until 1999, after which the use was terminated for eight years before it was re-introduced. Hydrogen peroxide was used between 1993 and 1997, terminated for eleven years and re-introduced in 2009. Diflubenzuron was not in use between 2001–2008 and teflubenzuron was not in use between 2002–2008. The pyrethroid

**Table 1. Consumption of medicines, 2000–2012.**

| Substance | 2000 | 2001 | 2002 | 2003 | 2004 | 2005 | 2006 | 2007 | 2008 | 2009 | 2010 | 2011 | 2012 |
|---|---|---|---|---|---|---|---|---|---|---|---|---|---|
| Azamethiphos | 0 | 0 | 0 | 0 | 0 | 0 | 0 | 0 | 66 | 1,884 | 3,346 | 2,437 | 4,059 |
| Hydrogen peroxide | 0 | 0 | 0 | 0 | 0 | 0 | 0 | 0 | 0 | 308,000 | 3,071,000 | 3,144,000 | 2,538,000 |
| Diflubenzuron | 12 | 0 | 0 | 0 | 0 | 0 | 0 | 0 | 0 | 1,413 | 1,839 | 704 | 1,611 |
| Teflubenzuron | 62 | 28 | 0 | 0 | 0 | 0 | 0 | 0 | 0 | 2,028 | 1,080 | 26 | 751 |
| Cypermethrin | 73 | 69 | 62 | 59 | 55 | 45 | 49 | 30 | 32 | 88 | 107 | 48 | 232 |
| Deltamethrin | 23 | 19 | 23 | 16 | 17 | 16 | 23 | 29 | 39 | 62 | 61 | 54 | 121 |
| Emamectin benzoate | 30 | 12 | 20 | 23 | 32 | 39 | 60 | 73 | 81 | 41 | 22 | 105 | 36 |

Overview of substance consumption given in kg active substance, years 2000–2012. Data from 1981–1999 are presented in S1 Table.

**Table 2. Consumption of medicines, 2013–2019.**

| Substance | 2013 | 2014 | 2015 | 2016 | 2017 | 2018 | 2019 |
|---|---|---|---|---|---|---|---|
| Azamethiphos | 3,037 | 4,630 | 3,904 | 1,269 | 204 | 160 | 154 |
| Hydrogen peroxide | 8,262,000 | 31,577,000 | 43,246,000 | 26,597,000 | 9,277,000 | 6,735,000 | 4,523,000 |
| Diflubenzuron | 3,264 | 5,016 | 5,896 | 4,824 | 1,803 | 378 | 1,296 |
| Teflubenzuron | 1,704 | 2,674 | 2,509 | 4,209 | 293 | 144 | 183 |
| Cypermethrin | 211 | 162 | 85 | 48 | 8 | 0 | 0 |
| Deltamethrin | 136 | 158 | 115 | 43 | 14 | 10 | 10 |
| Emamectin benzoate | 51 | 172 | 259 | 232 | 128 | 87 | 114 |

Overview of substance consumption given in kg active substance, years 2013–2019.

cypermethrin was used from 2000–2017, while the pyrethroid deltamethrin and the avermectin emamectin benzoate were used throughout the period 2000–2019.

As seen in Fig 1, the peak year for medicinal treatments was 2015, when 5.7 times as many tonnes of salmonids were treated compared to harvested–or phrased another way: each ton of

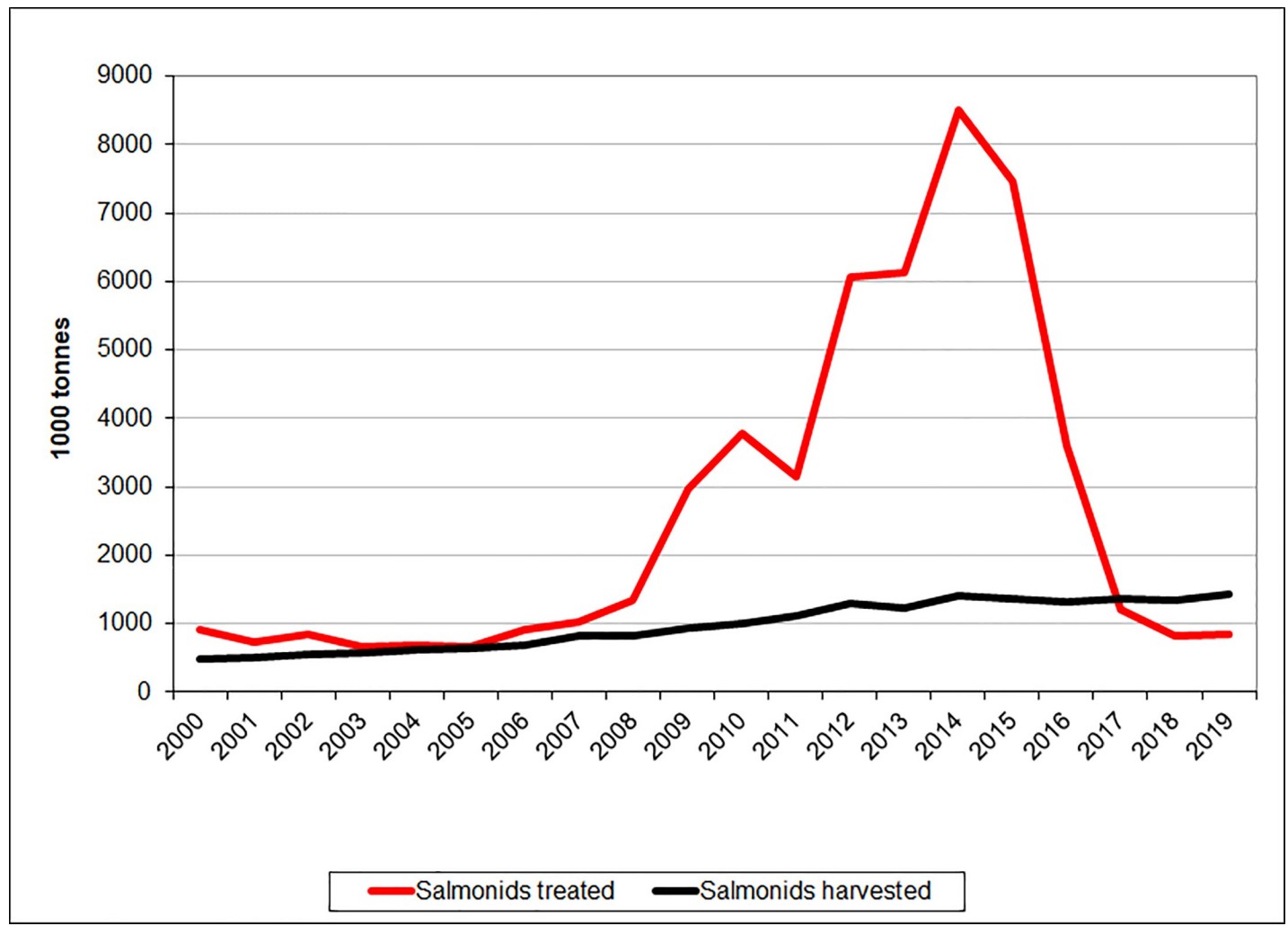

**Fig 1. Treated versus harvested salmonid volume.**

salmonids produced received on average more than five treatments that year, assuming that all treatments were conducted according to label. In the following three years, medicinal treatments drastically declined and in 2018 and 2019, only about half the volume of harvested salmonids had received medicinal treatments. The data are included in tabulated format in the S2 Table.

Presentation of the biomass of salmonids (in 1000 tonnes) harvested (black line) and treated for salmon lice with medicinal compounds (red line) in Norway from 2000–2019. The data are compiled from Statistics Norway (biomass harvested) and the Norwegian Institute of Public Health (use of medicinal products). The following assumptions have been made: 1) Bath treatments (azamethiphos, hydrogen peroxide, cypermethrin, deltamethrin) were conducted with a biomass density of 50 kg/m$^3$. 2) The dosages were: 0.1 g/m$^3$ azamethiphos; 0.015 g/m$^3$ cypermethrin; 0.002 g/m$^3$ deltamethrin; 1500 g/m$^3$ hydrogen peroxide; 0.042 g/kg diflubenzuron; 0.07 g/kg teflubenzuron; 0.00035 g/kg emamectin benzoate.

### Traditional bioassays

In Fig 2A, the EC$_{50}$ values in bioassays using deltamethrin from year 2000 to 2012 are shown. In the earliest bioassays (2000–2003), the EC$_{50}$ values were between 0 and 1 ppb. There is no available data in years 2004–2006, however in 2007 there are two observations of EC$_{50}$ = 0.7

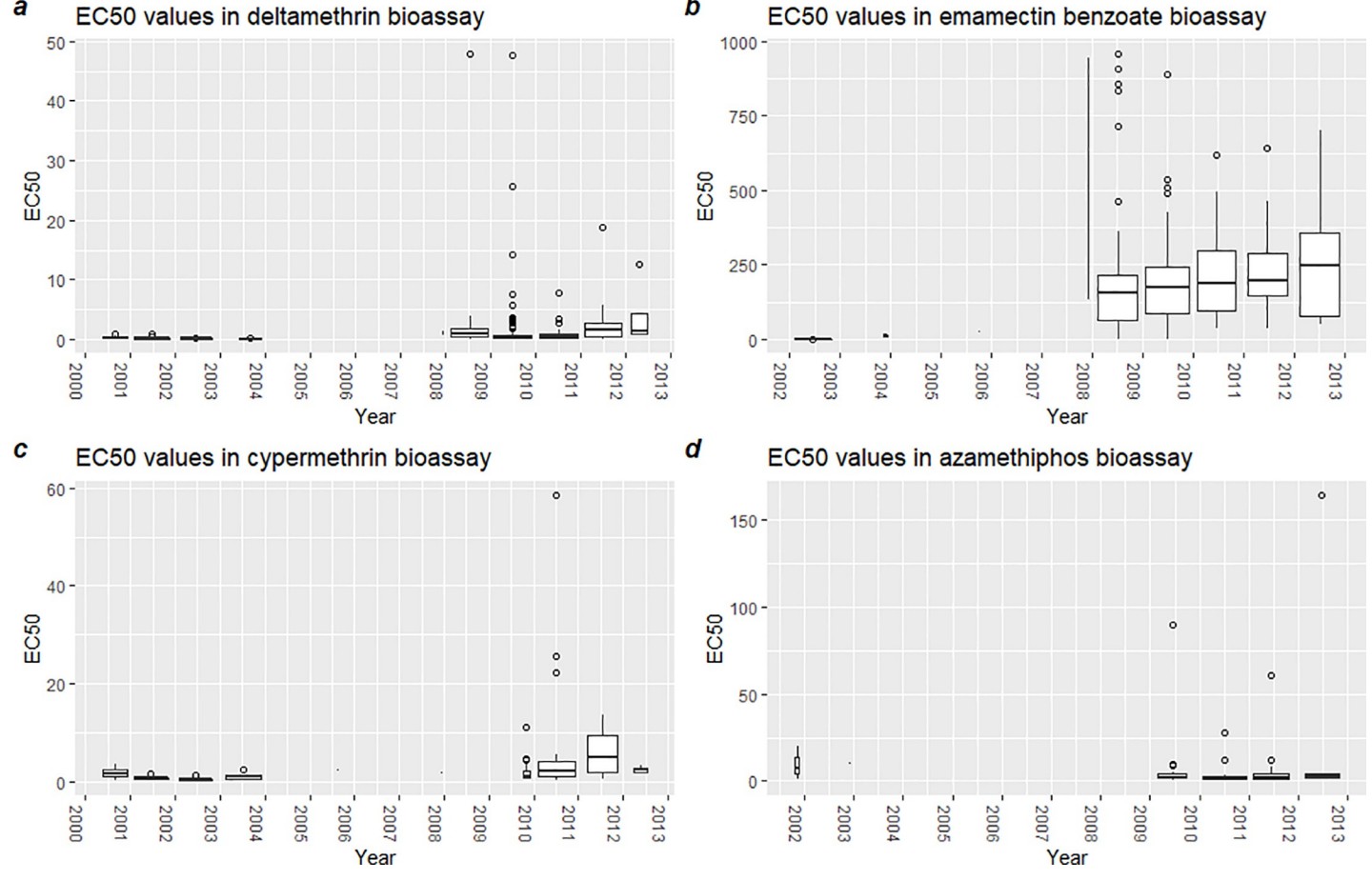

**Fig 2. Predicted EC$_{50}$ values in bioassays run in years 2000–2012.**

and 1.4 ppb. From 2008–2012, bioassays generally show $EC_{50}$ values of 0–7.5, with extreme values of up to around 140.

In Fig 2B, the $EC_{50}$ values in bioassays using emamectin benzoate from year 2002 to 2012 are shown (no data for 2000 and 2001). In the years 2002–2003, the $EC_{50}$ values ranged from 0.8–15.7 ppb. There are no available data from 2004. In 2005, the two available observations are $EC_{50}$ = 28.2 and 29.7 ppb. From 2007–2012, $EC_{50}$ values of up to 960 ppb were recorded. The median values were around 130–250 ppb.

In Fig 2C, the $EC_{50}$ values in bioassays using cypermethrin from year 2000 to 2012 are shown. In years 2000–2003, the $EC_{50}$ values ranged from 0.07 to 3.6 ppb. There are no available data from 2004, nor 2008. There was only one observation for 2005 and 2007 each, with $EC_{50}$ values of 2.4 and 1.8 ppb, respectively. From 2009–2012, the $EC_{50}$ values ranged from 0.04–145.6 ppb, with medians around 2.5–5 ppb.

In Fig 2D, the $EC_{50}$ values in bioassays using azamethiphos from year 2002 to 2012 are shown. There are no available data from year 2000. In years 2001–2002, the $EC_{50}$ values ranged from 0.8–20.7 ppb. There are no available data from the period 2003–2008. In the period 2009–2012, the $EC_{50}$ values ranged from 0.3 to 163.8 ppb. The median for the same period ranged from around 1–4 ppb. The data are also presented in a tabulated format with descriptive statistical parameters in the S3 Table.

Sensitivity ($EC_{50}$ values) in bioassays with (A) deltamethrin, (B) emamectin benzoate, (C) cypermethrin, (D) azamethiphos. Interquartile range (boxes), median value (bold line), largest and smallest value within 1.5 times interquartile range (whiskers) and outliers (dots) are given. For all substances, the variation in sensitivity increased substantially after 2008. A very limited number of assays were conducted in the years 2004–2007.

Common for all substances is that the $EC_{50}$ values (which indicates sensitivity level) are low and relatively consistent in years before 2007, while the range and number of extreme values increases drastically after 2008.

## Bioassays 2013–2019

In Fig 3A, the surviving proportion of the test population in bioassays using azamethiphos in years 2013–2019 are presented. As seen, the trend (as represented by the lowess curve in red) has been one of increasing survival until the end of year 2016. From 2017, the surviving proportion of the test population has declined steadily from around 0.63 (63%) to around 0.5 (50%) survival at the end of 2019—levels that haven't been measured since surveillance was commenced in 2013.

In Fig 3B, the surviving proportion of the test population in bioassays using deltamethrin in years 2013–2019 are presented. The same general trend is seen for these bioassays as for azamethiphos bioassays described above: survival increases steadily from just below 0.5 (50%) in late 2013 until a peak proportion of around 0.55 (55%) is reached at the end of 2016, after which it declines. The difference, however, is that the survival declines to levels below the start of the surveillance, at a proportion of about 0.37 (37%).

In Fig 3C, the surviving proportion of the test population in bioassays using emamectin benzoate in years 2014–2019 are presented. As seen, the survival in the bioassays increased from around 0.36 (36%) in 2014 to 0.63 (63%) in late 2016, after which it decreased to around 0.4 (40%) in late 2018. Unlike the remaining three substances, the surviving proportion of the test populations exposed to emamectin benzoate was higher in 2019 than in 2018, with a surviving proportion of 0.5 (50%).

In Fig 3D, the surviving proportion of the test population in bioassays using hydrogen peroxide in years 2014–2019 are presented. The figure shows that the survival increased from

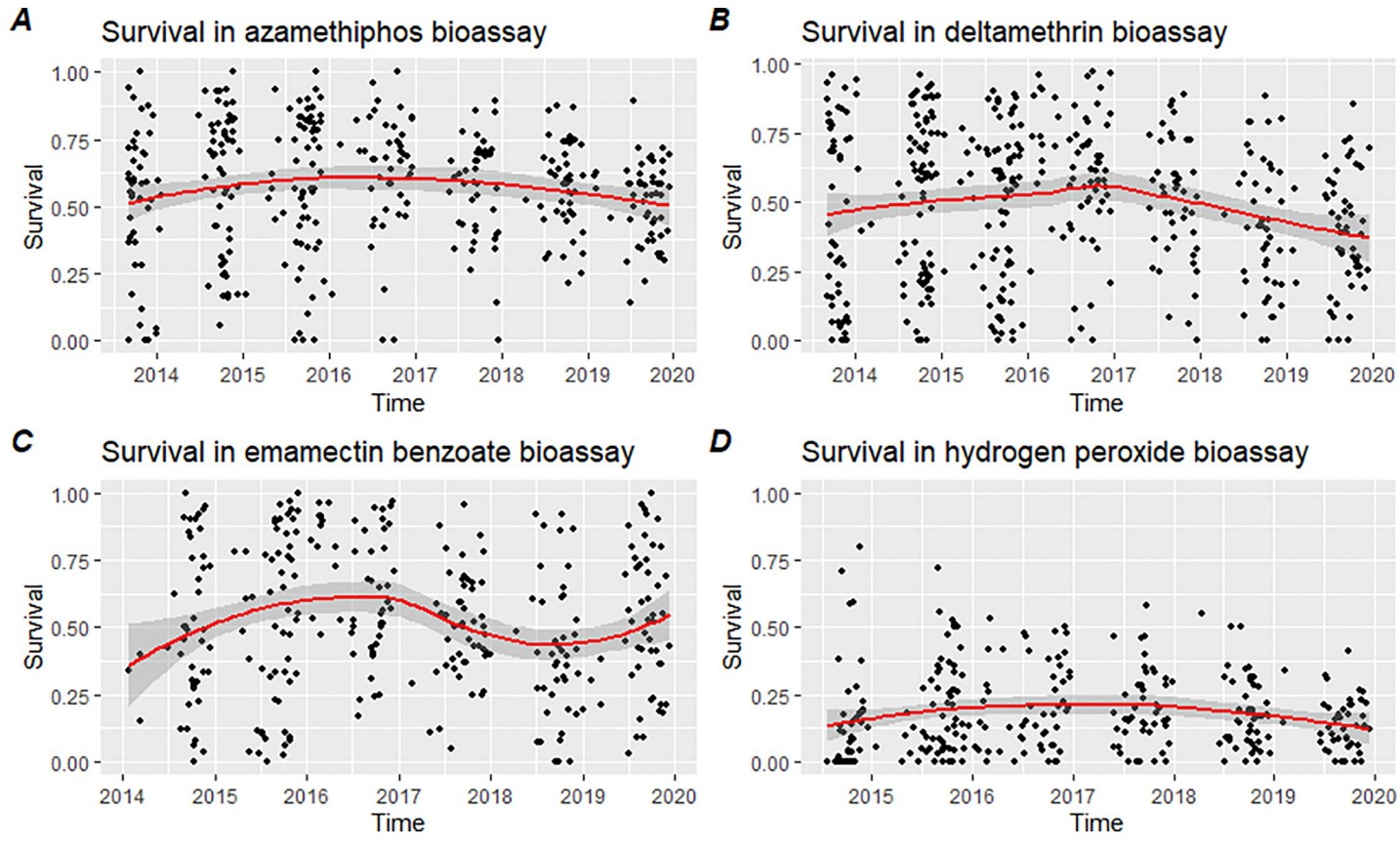

**Fig 3. Survival in bioassays in years 2013–2019.**

about 0.13 (13%) in late 2014 to right under 0.25 (25%) in 2016, where it remained until 2018 when the survival began to slowly decrease to 0.18 (18%). In 2019, the surviving proportion was at levels equaling those seen in 2014. The overall survival in the hydrogen peroxide bioassays was lower than for the other substances.

Proportion of test populations of salmon lice (*L. salmonis*) surviving bioassays with (A) 2 ppb azamethiphos, (B) 1 ppb deltamethrin, (C) 300 ppb emamectin benzoate or (D) 240 ppm hydrogen peroxide. The lowess curve that best fits the data (red line) and the 95% confidence intervals (gray area) are given. High survival rates indicate reduced sensitivity. For all substances, the survival rate peaked late 2016 to early 2017 and declined thereafter. For emamectin benzoate, it increased again in 2019.

Common for all substances is that a peak in the surviving proportion of the test population was recorded around late 2016 to early 2017 and followed by a decline in survival.

### Medicinal versus non-medicinal treatment frequencies

The annual frequencies of treatment events with medicinal or non-medicinal methods is presented in Fig 4. Medicinal treatments were unquestionably the dominant way of treating salmon between 2012 and 2015. 2015 was a peak year for medicinal treatments, with close to 1000 treatment events, i.e. weeks were a location has reported using a non-medicinal method, taking place. The number of non-medicinal methods stayed relatively constant these four years. In 2016, there was six-fold increase in the number of non-medicinal treatments conducted, while the number of medicinal treatments decreased somewhat. In

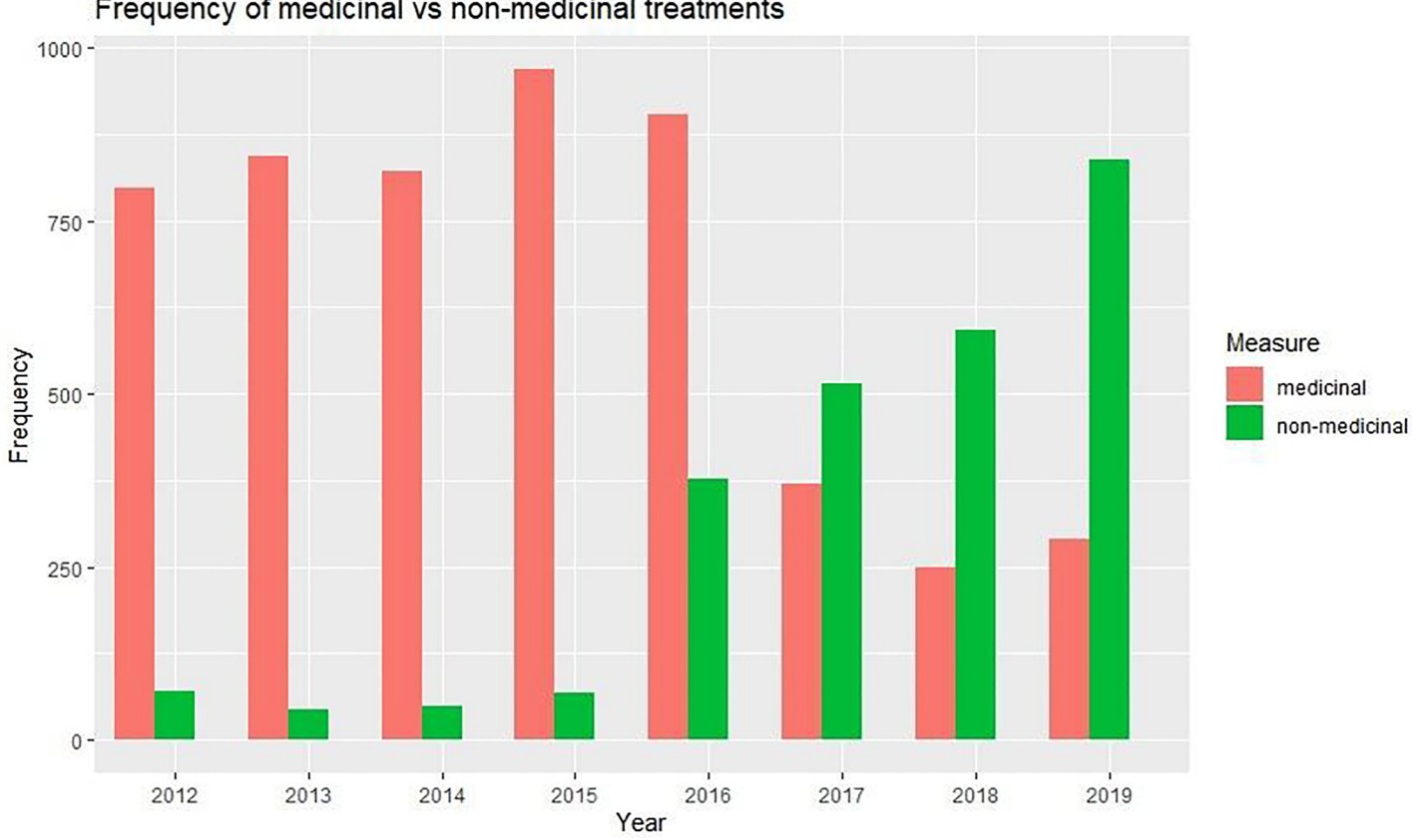

**Fig 4. Medicinal versus non-medicinal treatments per year.**

2017 non-medicinal methods became the most prevalent measure against salmon lice, with about 500 non-medicinal treatment events against 375 medicinal treatment events. In 2018, non-medicinal were almost three times as common as medicinal methods—and in 2019, around 3.3 non-medicinal treatments happened for every medicinal treatment event.

Annual frequencies (in number of events) of medicinal (red) and non-medicinal (green) treatments presented as a bar graph [20].

### Cleanerfish

As seen in Fig 5, the number of cleanerfish deployed were relatively stable in years 2000–2008, with a mean of 1,462.9 (x1000). Then from 2009–2017, the use of cleanerfish in Norwegian aquaculture increased dramatically. In 2018, the number decreased somewhat from the previous year while they increased and peaked in 2019 at 60,565 (x1000). The increase in deployment of cleanerfish coincided with the relative increase in the use of medicinal compounds against salmon lice (Fig 1) and reduction in sensitivity (Fig 2).

### Discussion

In this retrospective study we have sought to describe the measured resistance level in salmon lice (*L. salmonis*) along the Norwegian coast from 2000–2019 using all available sensitivity

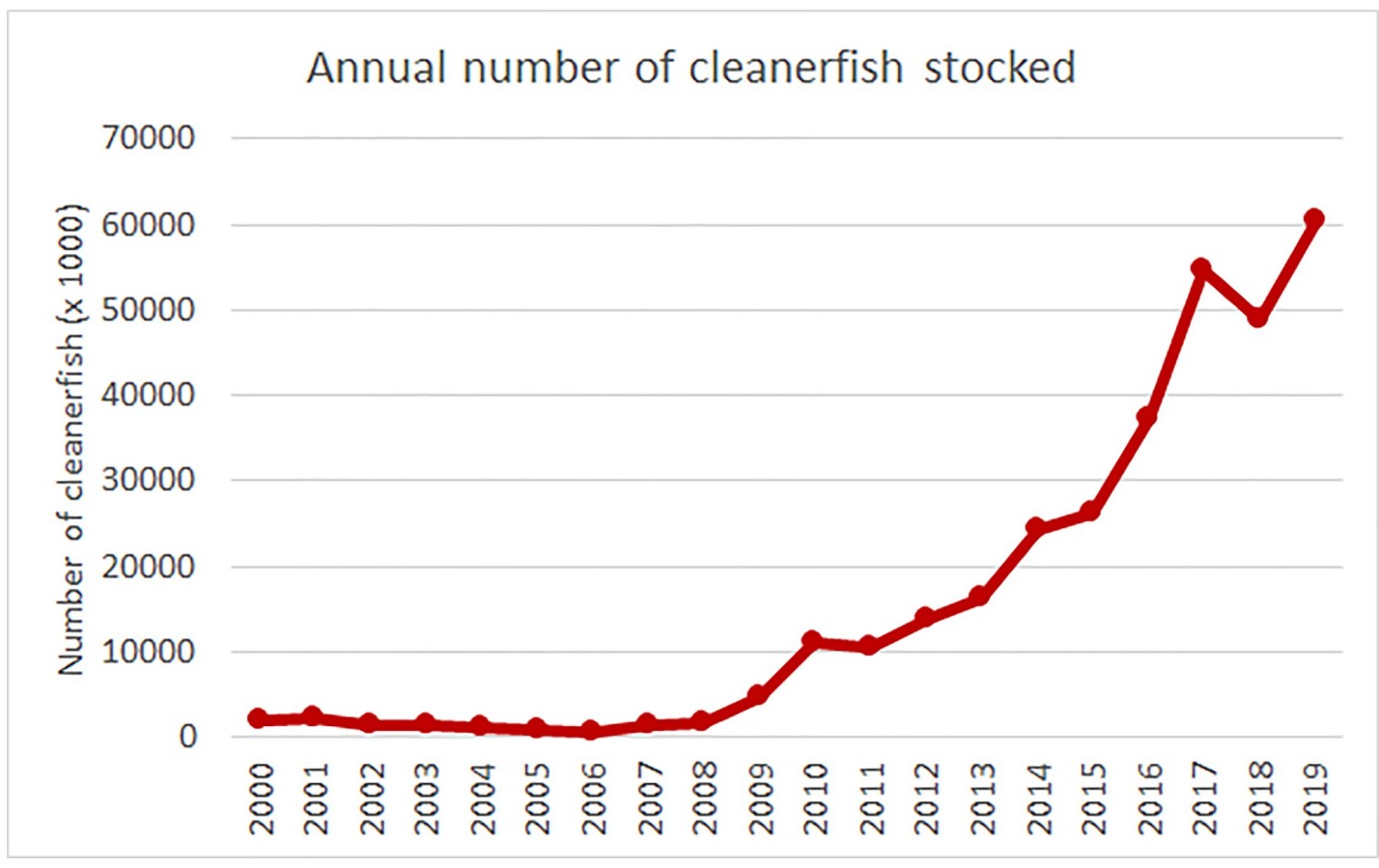

**Fig 5. Cleanerfish deployment per year.** Annual number of individual cleanerfish (in 1000) deployed in Norwegian aquaculture [21].

tests conducted in the period, combined with information about consumption of de-lousing substances and the emergence of non-medicinal techniques for lowering salmon lice infestations.

### Traditional bioassays (2000–2012)

The $EC_{50}$ values from 2000–2012 were derived from traditional bioassays with exposure time of 30–60 minutes and an observation time of 24 hours using six or more doses which were slightly changed throughout the time period. Hence, direct comparison of bioassays from these years and data from the standardized national surveillance program 2013–2019 is not possible. The dataset comprises–to our knowledge–all bioassays conducted in Norway in this period. Even though the sampling was partly selective and could introduce some bias, this is the only dataset describing emergence of reduced sensitivity to salmon lice treatments in the period 2000–2012. The data showed that the $EC_{50}$ values and the variation in sensitivity for the individual compounds was low between 2000 and 2003. As no treatment errors were reported in the period, they likely represent the natural sensitivity of the parasites. The average $EC_{50}$ values and the variation in results increased considerably from 2007 (emamectin benzoate), 2008 (deltamethrin) and 2009 (cypermethrin, azamethiphos). Only a few assays were conducted between 2004 and 2006, but all these were similar to results obtained in the first period. Thus, the serious resistance problems seem to have emerged in the last year(s) prior to 2008 when resistance was recognized as a major problem. This increase in variance is typical of beginning

resistance in populations, as most individuals will still be sensitive to the medicinal treatments taking place, while the resistant genotypes or biomarkers will become increasingly prevalent following the selective pressure that medicinal treatments constitute [24, 25].

These traditional bioassays indicated when the shift in sensitivity occurred but were not analyzed together with treatment efficacy data. The importance of establishing how $EC_{50}$ levels in bioassays compare to efficacy in field was raised by Helgesen & Horsberg [11]. In this study, clear correlations were found between bioassays and small-scale emulated treatments. This has also been touched upon in other studies [26, 27].

The sensitivity data generated through the resistance surveillance program funded by the Norwegian Food Safety Authorities were collected from 2013 for azamethiphos, deltamethrin and emamectin benzoate, and from 2014 for hydrogen peroxide. For the current study, the raw data from the program were used with permission from the Food Safety Authorities. Data from the surveillance program were not completely random as farms were chosen by local fish health services within predetermined geographical areas covering the entire coastline. The test protocol does not allow estimation of median effective concentrations ($EC_{50}$) and the results are therefore not directly comparable with the earlier results. The survival rate of parasites at the different concentrations tested was used as the indicator of the degree of sensitivity in the population. This can increase the level of uncertainty in the data, but was compensated for by the high number of assays conducted, in total 1486 over the period.

In a study focusing on the correlation between sea lice counts and emamectin benzoate treatments in Scotland, Lees et al. [28] demonstrated a gradual loss of efficacy over the years 2002–2006. At the same time, there was little focus on this fact since most treatments were successful. In the current study, reduced efficacy of emamectin benzoate could be demonstrated in Norway from 2007.

## Bioassays with two-doses, 2013–2019

The survival in two-dose bioassays with azamethiphos increased from 2014 to 2016 and then decline to 2014-levels in during 2018 (Fig 3). For deltamethrin, survival increased from 2014 to 2016. From 2016 the survival declined markedly to 2019. In bioassays with emamectin benzoate, survival increased from 2014 to 2016 and then declined. However, in 2019 the survival increased again to levels matching 2015 and 2017 levels. Lastly, the bioassays with hydrogen peroxide showed increase in survival from 2014 to 2017. From 2017 to 2019, the survival declined slowly.

Thus, survival in the bioassays increased until 2016 (2017 for hydrogen peroxide). Hydrogen peroxide was first taken on as an approved salmon louse treatment in 2009, which is later than the three other substances, and remained more efficient than the other compounds for å longer time due to a limited use the first years. The use increased substantially after 2012 (Tables 1 and 2).

These trends of decreasing resistance after a period of high resistance frequencies correlate well with the decline in the number of medicinal treatments from 2015 onward, as seen in Fig 1. When selection pressure from treatments decreases, or even disappears in some areas, resistance levels also decrease. This may in part be due to fitness costs associated with the resistance mechanisms, meaning that the processes in the salmon louse (or organism) yielding resistance are energy costly and hence become a disadvantage when exposure to the substance stops [29]. The effect of such fitness costs are difficult to measure, as it usually only amounts to a few percent [6, 29]. The discovery that survival in field bioassays has declined for all available chemicals used in lice control indicates that resistance has not become fully fixed in the population so that the absence of chemical pressure removes the advantage of carrying the resistance

mechanism. It is unlikely that resistance will disappear, however. The point mutation leading to organophosphate resistance in salmon lice, for example, was most likely present in the North Atlantic population well before treatments using organophosphates were initiated [30]. Being a naturally occurring mutation it will never disappear completely. Genes coding for resistance will be diluted in the population, but if the use of the compound increases again, they can be rapidly selected for [31–34]. This seems to be the case for emamectin benzoate, where the consumption declined from 2015 to 2018 (Table 2), reducing the selection pressure and subsequently leading to increased sensitivity levels (Fig 3C). From 2018 to 2019, the consumption increased again, and the sensitivity level decreased. It is important to note that the increase in sensitivity is moderate, and are in no way approaching the levels seen in the early 2000s.

The data presented in the current study do not reveal the precise mechanism that led to development of increased tolerance. Kaur and colleagues could infer a correlation between azamethiphos resistance and a single mutation in the gene coding for acetylcholin esterase [33, 34]. With regards to hydrogen peroxide resistance, Helgesen and colleagues discovered that resistant individuals had higher catalase activity compared to their $H_2O_2$ sensitive counterparts [35]. Pyrethroid resistance has in other arthropods mainly been associated with specific non-synonymous mutations in voltage-gated sodium channels, and such a mutation has been detected in the salmon louse [36]. However, maternal inheritance of resistance also points to mutations in the mitochondrial genome as a cause for resistance [37, 38]. For emamectin benzoate resistance, no conclusive mechanisms have been described.

Salmon lice are highly adaptable organisms, as the females have very high fecundity in favorable environments such as in the net pen of a salmon farm. They are sexually mature after only 52 days post hatching at 10˚C [39], each female may produce around 200 eggs per egg string and can produce up to 11 pair of egg strings within their life span [40]. Furthermore, the generation interval of salmon lice is relatively short and heritable advantages, such as medicinal resistance, has the potential of spreading very quickly if the selective pressure posed by treatments with the substance in question if upheld [25].

As seen in Fig 4, there has been a clear shift in de-lousing trends from almost exclusively medicinal from 2012 to 2015 to dominantly non-medicinal from 2017 and forward. The use of cleanerfish has also increased dramatically from the early 2000's until 2017, as seen in Fig 5. This shift can be explained by poor treatment efficacies from all available substances, which forces companies to either increase doses, repeat treatments with another medicinal compound or change strategy, i.e. choose non-medicinal methods. If lice levels are not controlled properly, the production company may lose their license. Also, companies using large quantities of chemicals in their production are often criticized for polluting the marine environment, affecting non-target species. Chemical in the fish meat is also an important concern among consumers. Lastly, a recent implemented regulation prevents use of medicinal delousing in areas closer than 500 meters from spawning or shrimp areas [41]. Thus, there are more reasons than resistance development motivating companies to change to non-medicinal control methods.

A selection pressure on the population can be exerted by any factor rendering a part of the population with increased chance of survival. As seen from Figs 4 and 5, the use of non-medicinal treatment alternatives like warm water, freshwater and mechanical removal of parasites have risen by 881.8% from 2015 to 2019. However, no reports exist to date regarding development of increased salmon lice tolerance towards these treatment options. The risk of salmon lice developing increased tolerance towards freshwater was recently reviewed [42]. For proactive reasons, objective bioassay tests are being developed [17], and a freshwater tolerance test was in 2019 included in the Norwegian resistance monitoring program [10].

The data compiled in this study demonstrated that resistance to chemical treatments started to evolve just after the mid 2000s (Fig 2) and led to a rapid increase in the use of medicinal compounds, which cumulated in 2014 (Fig 1, Tables 1 and 2). This again increased the resistance selection pressure and resistance level (Figs 2 and 3) until the efficacy of several compounds was almost lost in the mid 2010s. The industry then largely switched to non-medicinal treatments (Fig 4) and increased the deployment of cleanerfish (Fig 5), resulting in less selection pressure of medicinal compounds and a slight increase in sensitivity (Fig 3). The data also demonstrated that this slowly returning sensitivity easily can be jeopardized by a new increase in utilization of the medicinal compounds, as seen for emamectin benzoate between 2018 and 2019 (Fig 3C).

The sources used to extract the data underlying the current study are not without flaw. However, the advantage of the Norwegian transparency around these types of data has made it possible to compile information from several sources, and thus increase understanding and reliability. This should act as an encouragement to other salmonid producing countries to increase their transparency. In order to learn from past mistakes, an understanding of the events preceding for example resistance development in salmon lice is crucial.

## Conclusions

In this study we have offered explanations as to why the sensitivity of salmon lice along the Norwegian coast evolved as it did from 2000–2019, how the increase in tolerance coincides with increasing consumption of medicines in the period 2000–2015, and slowly increasing sensitivity when the consumption of medicines decreased. The need for alternatives to the limited number of delousing medicines available is further supported by the increase in number of cleanerfish deployed in the same period. This highlights the significance of avoiding heavy reliance on one or a few substance groups to combat ectoparasites, as it has proven to be a potent catalyst for resistance evolution.

This study further highlights the importance of transparency in the global industry. Norwegian aquaculture is unprecedented in its transparency with regards to operational details, which enables the industry to learn from poor choices in the past.

## Supporting information

**S1 Data. Relative treatment intensity calculations.**
(XLSX)

**S1 Table. Table of consumption of medicines in years 1981–1999.**
(XLSX)

**S2 Table. Sensitivity data (EC$_{50}$ values) from bioassays conducted in years 2000–2012.** Location identifying details have been anonymized.
(XLSX)

**S3 Table. Sensitivity data (surviving proportion of the population) from bioassays conducted in years 2013–2019.** Location identifying details have been anonymized.
(XLSX)

## Acknowledgments

The authors would like to thank all the production companies for giving their permission to publish data on salmon lice sensitivity measured in bioassays conducted before the national surveillance program was initiated.

## Author Contributions

**Conceptualization:** Tor Einar Horsberg, Kari Olli Helgesen.

**Data curation:** Elena Myhre Jensen, Tor Einar Horsberg, Sigmund Sevatdal, Kari Olli Helgesen.

**Formal analysis:** Elena Myhre Jensen, Sigmund Sevatdal, Kari Olli Helgesen.

**Funding acquisition:** Tor Einar Horsberg.

**Investigation:** Elena Myhre Jensen, Tor Einar Horsberg, Kari Olli Helgesen.

**Methodology:** Elena Myhre Jensen, Tor Einar Horsberg, Kari Olli Helgesen.

**Project administration:** Elena Myhre Jensen, Tor Einar Horsberg, Kari Olli Helgesen.

**Resources:** Elena Myhre Jensen, Tor Einar Horsberg.

**Supervision:** Tor Einar Horsberg, Kari Olli Helgesen.

**Validation:** Tor Einar Horsberg, Kari Olli Helgesen.

**Visualization:** Elena Myhre Jensen, Kari Olli Helgesen.

**Writing – original draft:** Elena Myhre Jensen.

**Writing – review & editing:** Elena Myhre Jensen, Tor Einar Horsberg, Sigmund Sevatdal, Kari Olli Helgesen.

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
