## [Decision Letter · Decision Letter 0]

6 Aug 2020

PONE-D-20-16251

Trends in de-lousing of Norwegian farmed salmon from 2000-2019 – consumption of medicines, salmon louse resistance and non-medicinal control methods

PLOS ONE

Dear Dr. Myhre Jensen,

Thank you for submitting your manuscript to PLOS ONE. After careful consideration, we feel that it has merit but does not fully meet PLOS ONE’s publication criteria as it currently stands. Therefore, we invite you to submit a revised version of the manuscript that addresses the points raised during the review process.

The paper is of interest for a wide audience of stakeholders and deserves publication. However, it needs a style revision according to the comments of the reviewers. Please consider all the reveiwers' comments during the manuscript revision.  

We look forward to receiving your revised manuscript.

Kind regards,

Aldo Corriero, Ph.D.

Academic Editor

PLOS ONE

Journal Requirements:

Reviewers' comments:

Reviewer's Responses to Questions

**Comments to the Author**

1. Is the manuscript technically sound, and do the data support the conclusions?

Reviewer #1: Partly

Reviewer #2: Yes

2. Has the statistical analysis been performed appropriately and rigorously? 

Reviewer #1: N/A

Reviewer #2: Yes

3. Have the authors made all data underlying the findings in their manuscript fully available?

Reviewer #1: Yes

Reviewer #2: Yes

4. Is the manuscript presented in an intelligible fashion and written in standard English?

Reviewer #1: Yes

Reviewer #2: Yes

5. Review Comments to the Author

Reviewer #1: Overall this is a worthwhile study and a significant quantity of work seems to have been done using archive data. I have made many small suggestions which at first sight, may appear to be major revision, but are mainly stylistic or done to improve understanding. Main concern is that the Discussion needs some redrafting (see below). I think there is a difference between the approach used compared to Overton et al 2019, but nevertheless it would be a good paper to read.

I wish you all the best with publication and hope the details below are of some assistance.

……………..

Abstract – overall would benefit from inserting some key data or figures from the overall paper.

L17 – Only Norway mentioned, of course the topic of the paper, however at the start of the Introduction mentioning that it is an international issue and associated global costs would increase the importance of the paper.

L20 – Norwegian Food Safety Authority (in capitals?)

Check referencing of websites and date accessed, especially in a paper that uses numbered references, do they need to be detailed at the end of the paper (?)

L52 and 76 – use abbreviations after first use alongside full names, then use abbreviations thereafter?

L63 – dominant

L64 – reference required / link to following reference

L78 – provide a specific year

L79 – in number…is this the application incidence, number of fish treated? Make this a little clearer.

L83 to 85 – I think Thermolicers are a technique rather than a brand name, perhaps use this here. “Lice lasers” may need to be explained a little more (and efficacy has been questioned recently, see Bui et al., 2020) so ensure the list of non-medicinal approaches is complete, correctly named and fully but briefly explained. Why are other approaches dismissed – is it because they are management tactics or physical separation methods that are not recorded or difficult to quantify?

L88 – “genetic variation in susceptibility, and hence survival and onward input into the gene pool” may be a better phrase to consider.

L90 – Product or medicine consumption, rather than “substance”?

………………

Methods – overall might benefit from a decision tree or table to show how data was collected and why some was discarded, currently there is quite a lot of text to work through.

Main difficulty on first reading, which reappears through the paper, is how the 2-dose system differs to the traditional bioassay and why this is treated differently from this point forward. I think this is eventually made clearer near the start of the discussion (see below) but there are some elements in the discussion which may be better suited earlier in the paper.

L101 – Provide precise web pages? Not that easy to find.

L138 and 139 – reason for excluding some of these data points needs clarification.

Statistics – is it possible or plausible to do any comparisons between years for any of this data (Fig 3 for example?) If the data resolution allows, is it possible to add a geographic element around Norwegian salmon farming zones (e.g. Kristoffersen et al., 2018)?

…………….

Results

Would Table 1 be easier to visualise as a line graph, and perhaps allow easier comparison with other time series data? If there is concern that the data arises from two different methods half-way through the time series, could this be physically split simply with a vertical line to denote the change?

L244 – 245….slaughtered (i.e. each ton of salmonids….year)…

Use of slaughtered in this section and Figure 1…does slaughtered=harvested? Word slightly suggests slaughtered=harvested + killed for welfare reasons, so make sure this is defined as harvested for food.

In particular, Fig 2 and Fig 3: Description of results – I think the text could be simplified a little bit, focusing on main trends, and potentially some information (e.g. missing years) placed in the legend. Figure 2 x-axis data (years) need to be slanted or made vertical, in horizonal format they are difficult to read.

Could figure 5 be superimposed onto figure 4, using a second x axis? Also, cleanerfish “deployed” into salmon pens rather than “stocked” (which doesn’t necessarily suggest a mixed species) may be a better word to use. A version of cleanerfish deployment in Norway, and research needs, appears in Powell et al., 2018 and this may be a good reference for the discussion.

L336 – “medicine was unquestionably…treating salmon”

L341 – remove “the scale tipped, and”

………….

Discussion

There seems to be much here, particularly in the first few paragraphs, that doesn’t naturally belong in this section. For instance, 380-385 is arguably better suited in the introduction or M&M, results/missing data for certain years are repeated again here in some detail, and any worries about bias could be mentioned, discussed or reassurances made in the Materials and Methods.

Typically, the important specifics and reasoning for the main trends appear first, with the section becoming more general and expanding to more general applications nearer the end. Removing some of the redundant text would allow more space for this.

PLOS ONE requirements need a “conclusion supported by data”, and since this work is not a standard review with plenty of references, nor a typical experimental paper, this needs to be made a lot clearer or supported with references (perhaps statistics?) for this phenomenological review of archive data (e.g. L452-455 needs references). For instance, the final conclusion suggests that there is "clear covariation" but one could argue this is subjectively analysed.

Also, perhaps some expansion of future work – e.g. explore any regional or geographical differences? Any way to make this data predictive or assist modeling? Will new medicines assist and does medicine testing need to be made more streamlined or is it too ecologically risky? Will there be impacts of some of the management tools or equipment approaches not recorded by the state? Are non-medicinal mechanisms infallible and could they become less efficient (e.g. some signs that cleanerfish are improving the survival of albino lice/carry disease, vs a list of methods required to improve efficiacy and sustainability; thermolicers recently questioned).

Reviewer #2: A very good manuscript, very well written and very helpful. It is also a very good example on how transparency on public data should be applied, alowing further transversal studies using historical series of data like this one. It is also a very comprehensive work and very relevant for the salmon aquaculture industry, showing the benefits of good practices.

Concerning the manuscript, I would like to recommend the authors to diferentiate in the introduction the 'control measures' /as they integrate both prevention/prophylaxis and treatment/therapeutics) than purely 'treatments'. In fact, the authors already differenciate both as they do not consider in this study measures such as fallowing, snorkels, tarpaulins to control plankton layer or deep feeding measures as they are preventive.

Concerning grammar, only two minor questions or typos.

line 88: genetical or genetic?

line 448: (a)

6. PLOS authors have the option to publish the peer review history of their article (what does this mean?). If published, this will include your full peer review and any attached files.

Reviewer #1: No

Reviewer #2: **Yes: **Francesc Padros

---

## [Author Response · Author response to Decision Letter 0]

29 Sep 2020

Comments from Reviewers and Editor Response from Authors

Reviewer # 1 

Overall this is a worthwhile study and a significant quantity of work seems to have been done using archive data. I have made many small suggestions which at first sight, may appear to be major revision, but are mainly stylistic or done to improve understanding. Main concern is that the Discussion needs some redrafting (see below). I think there is a difference between the approach used compared to Overton et al 2019, but nevertheless it would be a good paper to read. 

The Overton et al 2019 paper cited in the current manuscript is a paper describing clinical experiments with cold water treatment, while our manuscript can be classified as a retrospective epidemiological study. Thus, a different approach has been chosen. Another paper by Overton et al 2019 not cited (Salmon lice treatments and salmon mortality in Norwegian aquaculture: a review, https://doi.org/10.1111/raq.12299) reviewed some of the same sources as we did in the current manuscript, but did not present data regarding sensitivity of the parasites. We have now in the Introduction added a reference to this review with the following comment: 

" Overton et al. [14] compared reported mortality rates associated with medicinal and non-medicinal treatments and found that thermal operations caused greatest mortality increase, followed by mechanical treatments, hydrogen peroxide treatments and then treatments with azamethiphos, deltamethrin and cypermethrin."

Abstract – overall would benefit from inserting some key data or figures from the overall paper 

We have included some key figures in the abstract:

"The peak year for medicinal treatments was 2015, when 5.7 times as many tonnes of salmonids were treated compared to slaughtered." and

"Reduced sensitivity towards azamethiphos, deltamethrin and emamectin benzoate was evident from 2009, 2009 and 2008, respectively."

To comply with the requirement of max 300 words in the abstract, it has been slightly revised.

L17 – Only Norway mentioned, of course the topic of the paper, however at the start of the Introduction mentioning that it is an international issue and associated global costs would increase the importance of the paper. We have added / modified the first paragraph of the introduction as follows:

"According to the Intrafish Sea Lice Report 2019, annual costs associated with sea lice management were estimated at USD 525 million and USD 350 million in the 2 main markets, Norway and Chile [2]. In high numbers these parasites have the potential of critically wounding their salmonid hosts."

L20 – Norwegian Food Safety Authority (in capitals?) This authority is always presented with capital letters in official documents from the Norwegian government, e.g.: " The Norwegian Food Safety Authority is the government supervisory body for food safety." Thus, we have not made any changes

Check referencing of websites and date accessed, especially in a paper that uses numbered references, do they need to be detailed at the end of the paper (?) We have now detailed the websites cited as numbered references, including date accessed.

L52 and 76 – use abbreviations after first use alongside full names, then use abbreviations thereafter? L52: "Norwegian Food Safety Authority" is normally not abbreviated in English. When mentioned in the manuscript, the full name is used everywhere.

L76: In 2013, the institution NMBU was called the "Norwegian School of Veterinary Science", thus NMBU has been changed to that. No abbreviation necessary, as this is the only time it is mentioned. VESO is the Norwegian abbreviation for the "Veterinary Center for Contract Research", which has been added.

L63 – dominant "dominating" changed to "dominant"

L64 – reference required / link to following reference The reference has been added.

L78 – provide a specific year The sentence has been rephrased to "Since 2017, the number of treatments using non-medicinal methods of lowering infestation levels has overtaken the number of treatments using medicinal compounds [12]."

L79 – in number…is this the application incidence, number of fish treated? Make this a little clearer. This refers to the number of treatments, not the volume of biomass treated. This is now clarified.

L83 to 85 – I think Thermolicers are a technique rather than a brand name, perhaps use this here. “Lice lasers” may need to be explained a little more (and efficacy has been questioned recently, see Bui et al., 2020) so ensure the list of non-medicinal approaches is complete, correctly named and fully but briefly explained. Why are other approaches dismissed – is it because they are management tactics or physical separation methods that are not recorded or difficult to quantify? "Thermolicer" is the brand name for the equipment produced by Steinsvik AS. There is a similar type of equipment produced by Optimar AS called "Optilicer". We chose to use "warm water" and not brand names as the description of the technique since new types of equipment using the same principles may be developed in the future. 

We have added mechanical removal to the list. Here, there are also two types of equipment (brand names: FLS Avluser and SkaMik 1.5). The sentence has been changed to "These consist of fresh water bathing, warm water dips, cold water bathing [13], use of lasers to kill individual lice on the fish, mechanical removal of parasites by soft brushes and/or high pressure pumps and stocking cleaner fish with the farmed salmon." These are the non-medicinal treatments of manifest sea lice infestations used today. The other strategies mentioned are preventive strategies. We have added functional feed to this list. They are not dismissed, just omitted because they cannot be classified as "treatments".

L88 – “genetic variation in susceptibility, and hence survival and onward input into the gene pool” may be a better phrase to consider. We agree, and have changed the wording.

L90 – Product or medicine consumption, rather than “substance”? We have changed "substance" to "medicine".

Methods – overall might benefit from a decision tree or table to show how data was collected and why some was discarded, currently there is quite a lot of text to work through. We have not added such a figure, but tried to straighten up the text instead.

Main difficulty on first reading, which reappears through the paper, is how the 2-dose system differs to the traditional bioassay and why this is treated differently from this point forward. I think this is eventually made clearer near the start of the discussion (see below) but there are some elements in the discussion which may be better suited earlier in the paper. We have rewritten a paragraph in the Introduction to give a better background and explanation of the differences: " Resistant parasites are suspected when there is a loss of treatment efficacy, and verified through bioassays or other laboratory assays. Bioassays are most commonly used and are toxicological assays where groups of parasites are exposed to different concentrations of the medicinal compound. The sensitivity level is evaluated from the survival following exposure. Before 2013, laboratory six-dose bioassays developed by Sevatdal & Horsberg [8] were conducted by the Norwegian School of Veterinary Science, the Veterinary Center for Contract Research (VESO) and some fish health services. By analysis of the dose-response curve, the population sensitivity was evaluated by determination of the median effective concentration of the medicinal compound, the EC50 value. As a response to the emerging resistance, the Norwegian Food Safety Authority wanted to set up a coast-long surveillance program for resistance in Norway. The program was launched in 2013 [10]. The individual bioassays were to be conducted by local fish health services. It soon became clear that the six-dose assays were too complicated for field use on a larger scale. The bioassay protocols chosen were based on two-level bioassays plus a control group, where the lowest dose discriminates between fully sensitive parasites and parasites with reduced sensitivity, while the highest dose is predicting the treatment efficacy using the labelled dosage [11]. All available bioassay data, both six-dose assays and two-dose assays were compiled in connection with this study."

L101 – Provide precise web pages? Not that easy to find. A more precise web page has been included as reference [5] in the reference list. 

L138 and 139 – reason for excluding some of these data points needs clarification. Six-dose (traditional) bioassays conducted after 2012 were excluded because from 2013, the two-dose bioassays greatly outnumbered these. We originally wanted to use data from the overlapping years to compare results between these two protocols, but this was impossible since none of the assays were conducted on the same strains of parasites. A short explanation is added to the text: "Traditional six-dose bioassays after 2012 were excluded since the number of two-dose assays greatly outnumbered these from 2013 (n = 49)." 

Bioassays conducted with more than one compound (typically azamethiphos and deltamethrin in combination) were excluded, since it was not possible to evaluate to which of the compound the parasite had developed reduced sensitivity. The text now reads: "Bioassays in which a combination of two (or more) substances were used in the same assay were excluded as the sensitivity level towards the individual compounds could not be determined (n = 12)."

Another clarification: " One bioassay with only adult males was excluded (n = 1)"

Statistics – is it possible or plausible to do any comparisons between years for any of this data (Fig 3 for example?) If the data resolution allows, is it possible to add a geographic element around Norwegian salmon farming zones (e.g. Kristoffersen et al., 2018)? We have certainly tried this. The problem is that the variation between individual observations each year is so big that statistical tests like ANOVA fail to detect differences. Some significant differences can be detected between some years for some substances, but it is only when looking at all data from the whole country combined (Fig. 3) that the overall trends stand out. We therefore decided only to present the lowess curve with 95 % CI for all data.

When we tried to split the data into the 13 regulatory regions, the number of observations per region became too small to get meaningful results. Even by dividing the coastline into North -, Mid - and South Norway, the data per region were quite blurred because there are big variations within each of these regions. Thus, we have not included the statistics and figures from these analyses.

Results

Would Table 1 be easier to visualise as a line graph, and perhaps allow easier comparison with other time series data? If there is concern that the data arises from two different methods half-way through the time series, could this be physically split simply with a vertical line to denote the change? Table 1 and 2, "Consumption of medicines, 2000-2012" and "...2013-2019", is in our opinion not suited for visualization as a line graph. Due to the huge difference in potency of individual compounds, the number could only be fitted using a logarithmic Y-axis. It will be less informative (see below), and the precise data will be lost (although they could be presented as supplementary data). We prefer to have these data presented as a table and have not made any changes.

L244 – 245….slaughtered (i.e. each ton of salmonids….year)…

Use of slaughtered in this section and Figure 1…does slaughtered=harvested? Word slightly suggests slaughtered=harvested + killed for welfare reasons, so make sure this is defined as harvested for food. The reviewer is correct, the meaning is "harvested"; the data is pulled from Statistics Norway where the Norwegian word "slaktet" is used for "harvested". We have changed the wording throughout the MS.

In particular, Fig 2 and Fig 3: Description of results – I think the text could be simplified a little bit, focusing on main trends, and potentially some information (e.g. missing years) placed in the legend. Figure 2 x-axis data (years) need to be slanted or made vertical, in horizonal format they are difficult to read. The figure texts have been simplified:

Fig. 2: "Sensitivity (EC50 values) in bioassays with (A) deltamethrin, (B) emamectin benzoate, (C) cypermethrin, (D) azamethiphos. Interquartile range (boxes), median value (bold line), largest and smallest value within 1.5 times interquartile range (whiskers) and outliers (dots) are given. For all substances, the variation in sensitivity increased substantially after 2008. A very limited number of assays were conducted in the years 2004 - 2007."

Fig. 3: " Proportion of test populations of salmon lice (L. salmonis) surviving bioassays with (A) 2 ppb azamethiphos, (B) 1 ppb deltamethrin, (C) 300 ppb emamectin benzoate or (D) 240 ppm hydrogen peroxide. The lowess curve that best fits the data (red line) and the 95% confidence intervals (gray area) are given. High survival rates indicate reduced sensitivity. For all substances, the survival rate peaked late 2016 to early 2017 and declined thereafter. For emamectin benzoate, it increased again in 2019."

Fig. 2 X-axis has been set vertically

Could figure 5 be superimposed onto figure 4, using a second x axis? Also, cleanerfish “deployed” into salmon pens rather than “stocked” (which doesn’t necessarily suggest a mixed species) may be a better word to use. A version of cleanerfish deployment in Norway, and research needs, appears in Powell et al., 2018 and this may be a good reference for the discussion. We agree that "deployment" is a better word than "stocking" and have changed this throughout the manuscript.

Information about deployment of cleanerfish between 2000 and 2011 will be lost if the line graph (Fig 5) is superimposed on the bar chart (Fig 4), since the Barentswatch data (Fig 4) only are available from 2012 and onwards. From Fig 5, it is evident that the deployment of cleanerfish increased substantially from 2009 when resistance to medicinal compounds started to emerge. This is now commented in the text: 

"The increase in deployment of cleaner fish coincided with the relative increase in the use of medicinal compounds against salmon lice (Fig 1) and reduction in sensitivity (Fig 2)."

L336 – “medicine was unquestionably…treating salmon” The sentence has been changed to " Medicinal treatments were unquestionably the dominant way of treating salmon between 2012 and 2015."

L341 – remove “the scale tipped, and” Done

Discussion

There seems to be much here, particularly in the first few paragraphs, that doesn’t naturally belong in this section. For instance, 380-385 is arguably better suited in the introduction or M&M, results/missing data for certain years are repeated again here in some detail, and any worries about bias could be mentioned, discussed or reassurances made in the Materials and Methods. Typically, the important specifics and reasoning for the main trends appear first, with the section becoming more general and expanding to more general applications nearer the end. Removing some of the redundant text would allow more space for this.

 We agree that parts of the Discussion are unnecessary repetitions of facts stated earlier. We have tried to minimize this in the revision. We have also moved and rewritten some parts, e.g. the section mentioned (lines 380-385) to the Introduction. Re the organization of the Discussion, we do not fully agree. We think that we have kept reasonably well to the usual organisation with 1) discussion of own results, and 2) more general discussion. However, as this MS describes data compiled from a variety of sources, the MS would appear very unstructured if all datasets were to be discussed twice, first specifically and then from a general point of view. We have though at the end added a paragraph with an overall discussion that is supported across the datasets: " The data compiled in this study demonstrated that resistance to chemical treatments started to evolve just after the mid 2000s (Fig 2) and led to a rapid increase in the use of medicinal compounds, which cumulated in 2014 (Fig 1, Table 1 and 2). This again increased the resistance selection pressure and resistance level (Figs 2 and 3) until the efficacy of several compounds was almost lost in the mid 2010s. The industry then largely switched to non-medicinal treatments (Fig 4) and increased the deployment of cleanerfish (Fig 5), resulting in less selection pressure of medicinal compounds and a slight increase in sensitivity (Fig 3). The data also demonstrated that this slowly returning sensitivity easily can be jeopardized by a new increase in utilization of the medicinal compounds, as seen for emamectin benzoate between 2018 and 2019 (Fig 3c)."

PLOS ONE requirements need a “conclusion supported by data”, and since this work is not a standard review with plenty of references, nor a typical experimental paper, this needs to be made a lot clearer or supported with references (perhaps statistics?) for this phenomenological review of archive data (e.g. L452-455 needs references). For instance, the final conclusion suggests that there is "clear covariation" but one could argue this is subjectively analysed. We are not quite sure what the point is here. Although we have used some publicly available data in this study, we have also presented new, unpublished data on resistance development between 2000 and 2012. Also, some of the publicly available data are remodelled (bioassays 2013 - 2019) and put in context with other public data sources. Thus, we do not consider this study to be a review. We do agree that more statistical analyses may have strengthened some of the conclusions, especially covariation between the parameters. We have changed "clear covariation" to "coincides". This study is mainly a descriptive study, which we now have highlighted in the aims: "In this descriptive retrospective study, ..." 

L452-455: This was referenced, but the references came after the next sentence. Reference [29] addresses the point made in these lines directly and has been copied directly after. 

Also, perhaps some expansion of future work – e.g. explore any regional or geographical differences? Any way to make this data predictive or assist modeling? Will new medicines assist and does medicine testing need to be made more streamlined or is it too ecologically risky? Will there be impacts of some of the management tools or equipment approaches not recorded by the state? Are non-medicinal mechanisms infallible and could they become less efficient (e.g. some signs that cleanerfish are improving the survival of albino lice/carry disease, vs a list of methods required to improve efficiacy and sustainability; thermolicers recently questioned). We agree that it would have been nice to be able to study regional differences. But as explained earlier, this led to the number of observations in some of the areas becoming too low for any sensible evaluation. Regarding non-medicinal treatment becoming less efficient with time, this is certainly a concern. We had discussed it in the paper ("A selection pressure on the population can be exerted by any factor rendering a part of the population with increased chance of survival. ..."), and this is also the focus of a separate research project. Regarding welfare implications of non-medicinal treatments, thus has also been discussed: "Although safer for the environment and no detected resistance development so far, some of these methods involve stressful crowding, pumping and other types of handling. Their impact on fish welfare have therefore been questioned. ..."

Reviewer #2 

A very good manuscript, very well written and very helpful. It is also a very good example on how transparency on public data should be applied, alowing further transversal studies using historical series of data like this one. It is also a very comprehensive work and very relevant for the salmon aquaculture industry, showing the benefits of good practices. Thank you.

Concerning the manuscript, I would like to recommend the authors to diferentiate in the introduction the 'control measures' /as they integrate both prevention/prophylaxis and treatment/therapeutics) than purely 'treatments'. In fact, the authors already differenciate both as they do not consider in this study measures such as fallowing, snorkels, tarpaulins to control plankton layer or deep feeding measures as they are preventive. We agree with this comment. We have now distinguished between non-medicinal treatment methods and preventive strategies, which we have only mentioned, but not discussed: " Preventive strategies such as synchronized fallow periods within production zones, synchronization of treatments, use of snorkel cages, functional feed, deep water feeding and plankton nets will not be addressed further in this study."

---

## [Editor Report · Decision Letter 1]

6 Oct 2020

Trends in de-lousing of Norwegian farmed salmon from 2000-2019 – consumption of medicines, salmon louse resistance and non-medicinal control methods

PONE-D-20-16251R1

Dear Dr. Myhre Jensen,

We’re pleased to inform you that your manuscript has been judged scientifically suitable for publication and will be formally accepted for publication once it meets all outstanding technical requirements.

Kind regards,

Aldo Corriero, Ph.D.

Academic Editor

PLOS ONE

Additional Editor Comments (optional):

All the comments of the two reviewers have been taken into consideration in the revised manuscript that can now be accepted for publication.
---

## [Editor Report · Acceptance letter]

20 Oct 2020

PONE-D-20-16251R1 

Trends in de-lousing of Norwegian farmed salmon from 2000-2019 – consumption of medicines, salmon louse resistance and non-medicinal control methods 

Dear Dr. Myhre Jensen:

I'm pleased to inform you that your manuscript has been deemed suitable for publication in PLOS ONE. Congratulations! Your manuscript is now with our production department. 

Kind regards, 

on behalf of

Dr. Aldo Corriero 

Academic Editor

PLOS ONE